# Effect of household relocation on child vaccination and health service utilisation in Dhaka, Bangladesh: a cross-sectional community survey

Lily Horng,[1] Nadira Sultana Kakoly,[2] Jaynal Abedin,[2] Stephen P Luby[1,2]

[1]Infectious Diseases and Geographic Medicine, Stanford University, Stanford, California, USA
[2]International Centre for Diarrhoeal Disease Research, Bangladesh (icddr,b), Dhaka, Bangladesh

**Correspondence to**
Dr Lily Horng;
lhorng@stanford.edu

## ABSTRACT

**Objective** To explore the relationship between household relocation and use of vaccination and health services for severe acute respiratory illness (ARI) among children in Dhaka, Bangladesh.

**Design** Analysis of cross-sectional community survey data from a prior study examining the impact of *Haemophilus influenzae* type b vaccine introduction in 2009 on meningitis incidence in Bangladesh.

**Setting** Communities surrounding two large paediatric hospitals in Dhaka, Bangladesh.

**Participants** Households with children under 5 years old who either recently relocated $\leq$12 months or who were residentially stable living $\geq$24 months in their current residence (total n=10 020) were selected for this study.

**Primary outcome measures** Full vaccination coverage among children aged 9–59 months and visits to a qualified medical provider for severe ARI among children under 5 years old.

**Results** Using vaccination cards with maternal recall, full vaccination was 80% among recently relocated children (n=3795) and 85% among residentially stable children (n=4713; $\chi^2$=37.2, p<0.001). Among children with ARI in the prior year, 69% of recently relocated children (n=695) had visited a qualified medical provider compared with 82% of residentially stable children (n=763; $\chi^2$=31.9, p<0.001). After adjusting for demographic and socioeconomic characteristics, recently relocated children were less likely to be fully vaccinated (prevalence ratio [PR] 0.97; 95% CI 0.95 to 0.99; p=0.016) and to have visited a qualified medical provider for ARI (PR 0.88; 95% CI 0.84 to 0.93; p<0.001).

**Conclusions** Children in recently relocated households in Dhaka, Bangladesh, have decreased use of vaccination and qualified health services for severe ARI.

## INTRODUCTION

Pneumonia or acute respiratory illness (ARI) is the leading cause of death globally in children under 5 years old and lower respiratory tract infections caused an estimated 652 572 child deaths in 2016.[1] Many causes of ARI are preventable by vaccines such as *Streptococcus pneumoniae* (attributed to 52% of global pneumonia child deaths) and *Haemophilus*

## Strengths and limitations of this study

► This study examined a rich dataset from prior community surveys in Dhaka, Bangladesh, to explore associations between household relocation and utilisation of vaccination and qualified child health services.

► Vaccination was evaluated using different measurements and age ranges to explore trends in the relationship between mobility and vaccination.

► Effect of household relocation on use of child health services was found even after adjusting for socioeconomic factors known to impact health-seeking behaviour.

► Limitations include lack of detailed data on mobility patterns and costs of health services.

*influenzae* type b (Hib) (7% of deaths).[1] The majority of the world's population now lives in urban areas and this population is expected to grow from 54% in 2014 to 66% in 2050, an estimated increase of 2.5 billion people.[2 3] With this increase, 90% of growth is projected in Asia and Africa.[3] The population living in urban slums is also expected to increase from 881 million in 2014 to 2 billion in 2030, in large part due to rural-urban migration.[2 4]

In many low- and middle-income countries, vaccination and childhood mortality rates among urban poor are worse than among other urban groups and even rural populations.[5 6] In addition, residents of slums have poor health outcomes due to lack of reliable access to housing, clean water, sanitation, education and health services.[4 5 7–9] In Nigeria, a 2010 study examining 2003 Demographic and Health Survey data of 6029 children 12 months and older found full immunisation among 24.3% of rural non-migrant, 15.2% of urban non-migrant and 8.5% of rural-urban migrant children.[10] In Bangladesh, a comparison study of the 2013 Urban Health Survey and 2014 Demographic and Health

Survey found under-5 child mortality rates of 46 per 1000 livebirths nationally, 41 in Dhaka, 49 in rural areas and 57 in urban slums.[5] One recent systematic review found community factors associated with vaccination coverage in the urban poor included socioeconomic characteristics, vaccination knowledge and beliefs, access to care and recent rural-urban migration.[6]

Residential mobility has been recognised as an important contributor to healthcare use in high-income countries, with relocation associated with decreased use of preventive and curative services.[11 12] One study using a 1998 US national health survey found that duration, distance and frequency of moving were all predictors of decreased use of child health services even after accounting for sociodemographic factors. Households who had moved within 12 months accessed fewer preventive child health services compared with households living in their current residence over 36 months (OR 3.1, 95% CI 2.5 to 3.7).[12] Recently relocated households also accessed fewer curative services (OR 3.3, 95% CI 2.6 to 4.2).[12] Frequent moving also impacted children's long-term cognitive function and behavioural problems into adulthood.[13] Moving can impact health through various social determinants. In one conceptual framework by the WHO, structural determinants of social, economical and political contexts influence intermediary determinants of material, behavioural, psychosocial and health system factors that ultimately shape an individual's health.[14] Relocation can improve an individual's socioeconomic position in the long-term, but mobility often disrupts material resources, psychosocial support and healthcare access.

Few studies examine mobility and healthcare utilisation in low- and middle-income countries despite high population relevance: approximately 43% of urban residents in middle-income countries and 78% in low-income countries live in slums.[3 4 6 10 15 16] In the 2010 Nigeria study, urban non-migrant children had 1.7 times higher odds of being fully immunised than rural-urban migrants (univariate OR 1.67, 95% CI 1.20 to 2.32).[10] This association between migration and immunisation was independent of demographic factors, but was attenuated and partially explained by socioeconomic characteristics and maternal healthcare utilisation in multivariable analyses.[10] In this Nigeria study, migrant status was defined as moving within 10 years.[10] A 2010 cross-sectional survey in India examined 746 rural-urban migrant mothers with children under 2 years old: 339 were 'recent' migrants who moved to Delhi within 5 years and 407 were 'settled' migrants in Delhi for at least 5 years.[16] For age-appropriate children, 81% of settled migrants and 64% of recent migrants were fully immunised per national guidelines.[16] Settled migrant children had 1.9 times higher odds of being fully immunised than recent migrants after adjusting for demographics, socioeconomics and maternal healthcare utilisation (adjusted OR 1.93, 95% CI 1.18 to 3.14).[16]

Studying urban health services in Bangladesh is useful because Bangladesh is the world's most densely populated country that is not a city-state: the population of the capital Dhaka will increase from an estimated 16 to 27 million by 2030.[3 17] Furthermore, the government has a strong national Expanded Programme on Immunisation and active health systems research.[17 18] In 2011, full vaccination rates among children age 12-23 months were 80% nationally in Bangladesh, 75% in Dhaka, but only 43% to 67% in Dhaka slums.[17–21] Prior studies found that household turnover was as high as 50% in 1 year, comprehensive provider-led vaccination interventions were effective, but too expensive to sustain and street children were very hard to reach with interventions in Dhaka.[19 22 23]

To explore the relationship between residential mobility and healthcare utilisation in Dhaka, we used data from a study showing Hib vaccine introduction into Bangladesh's Expanded Programme on Immunisation in 2009 dramatically reduced rates of Hib meningitis and purulent meningitis in children.[24] We conducted secondary analysis of the Hib impact study's community survey data to determine whether recently relocated children were: (1) less likely to be fully vaccinated per Expanded Programme on Immunisation guidelines and (2) less likely to use qualified health services for severe acute respiratory illness than residentially stable children.

## METHODS
### Study design and setting
Hib conjugate vaccine was introduced into Bangladesh's Expanded Programme on Immunisation in 2009 and the Hib impact study conducted pre and post-vaccine surveillance of meningitis in children under 5 years old using hospital records and community surveys surrounding two large paediatric hospitals in Dhaka: Dhaka Shishu and Shishu Shastya Foundation Hospital.[24] Field researchers consecutively enrolled 100 children discharged with a diagnosis of meningitis and/or encephalitis from the two study hospitals, visited households and recorded household geographical positional system coordinates. The catchment area was defined as the area containing >80% of households with children discharged with meningitis and within 1 hour of transport to either hospital. Field teams divided the catchment area into 1748 equal-sized rectangles and randomly selected 100 rectangles as clusters. Teams surveyed each household with a child under 5 years old within 98 clusters. Two clusters were within a military cantonment, thus inaccessible. Households were asked about: (1) routine vaccinations using vaccination cards and maternal recall and (2) healthcare use for children with illnesses in the prior 12 months suggestive of meningoencephalitis defined as: any serious illness with acute onset of fever with either convulsions or unconsciousness or altered mental status.[24] Data were collected 1 year before (2008) and after (2010) Hib vaccine introduction.

## Study population

We used the Hib impact study's pre-vaccine community surveillance data and included children based on mobility status: (1) children living in their current residence ≤12 months who we classified as 'recently relocated' and (2) children living ≥24 months in their current residence who we classified as 'residentially stable'. This definition of mobility/migration status has been used in prior studies.[6 12] We excluded children living in their current residence for 13-23 months who we classified as 'intermediately mobile'.

## Study outcomes

Our two primary outcomes focused on healthcare utilisation: (1) full vaccination among children aged 9-59 months and (2) visit to a qualified medical provider among children under 5 years old who had severe ARI symptoms within the prior 12 months. We defined full vaccination per Bangladesh Expanded Programme on Immunisation guidelines in 2008 (before Hib vaccine): one dose of BCG vaccine against tuberculosis; three doses of combined vaccine against diphtheria, pertussis and tetanus; three doses of oral polio vaccine (excluding polio vaccine given at birth) and one dose of measles vaccine. Government guidelines recommended children to receive all these vaccinations before 9 months of age. Any doses of pentavalent vaccine, which includes diphtheria, pertussis, tetanus, hepatitis B and Hib, were included in vaccination analyses. We defined severe ARI as cough or difficulty breathing plus any danger sign: stridor, chest in-drawing, difficulty drinking/breastfeeding, vomiting, cyanosis, convulsions, lethargy or unconsciousness. We defined a qualified medical provider as having a Bachelor of Medicine degree or higher.

## Data analysis

We compared sociodemographic and health characteristics between residentially stable and recently relocated households. For continuous variables, we calculated means with standard errors and t-tests adjusting for cluster. For categorical variables, we calculated percentages and $\chi^2$-tests. To construct a wealth index, we used polychoric principal components analysis including: housing (number of rooms; free, rental or owned housing; main material of roof, walls and floors), cooking fuel, drinking water, sanitation and mobile phone ownership.[25–27] We then divided households into wealth quintiles. We did not include durable assets such as furniture items because ownership of these goods could be associated with duration of residency.

To examine the magnitude of association between mobility and study outcomes of vaccination and visit to a qualified medical provider for severe ARI, we used modified Poisson regression with robust cluster variance to estimate prevalence ratios (PRs).[28 29] We chose modified Poisson regression to model PRs for common binary outcomes because logistic regression is more applicable to rare outcomes and because log-binomial regression

models may fail to converge. We conducted univariate analyses to estimate individual effects of mobility, demographics, socioeconomics and health services knowledge (ie, knowledge of local hospital) on healthcare utilisation. Missing data regarding main study outcome of ARI were handled through listwise deletion. Given large number of missing vaccination cards, we analysed vaccination in two ways: (1) using vaccination cards plus maternal recall and (2) using vaccination cards alone. We conducted multivariable analyses examining the association between mobility and healthcare utilisation, adjusting for demographics and socioeconomics known to influence health-seeking behaviour.[9 16 30–32] Regression diagnostics included checks for influential observations with Cook's distance calculations.

## Ethics

The Ethical Review Committee of the International Centre for Diarrhoeal Disease Research, Bangladesh (icddr,b), reviewed and approved the study protocol. Written informed consent was obtained from all participants before taking part in the initial Hib impact study.

## Patient and public involvement

No participants were directly involved in development of the research questions and outcomes. No participants were involved in the design or conduct of the study. There are no plans to disseminate the results of the research individually to study participants.

## RESULTS

We surveyed a total of 10 720 households with children less than 5 years old: 42% of households had recently relocated within 12 months, 51% were residentially stable living in their current residence over 24 months and 7% were intermediately mobile (figure 1). We excluded from subsequent analyses 700 children living in their current residence 13-23 months and classified as intermediately mobile. For the healthcare utilisation analysis, 1458 children had severe ARI symptoms within the 12 months prior to survey. For the vaccination analysis, 8508 children were age 9-59 months and thus should have completed all Expanded Programme on Immunisation-recommended vaccinations. Household demographics, parental education, occupation and hospital knowledge were available for all households. Missing data included: income for 12 households, meningitis symptoms for three children, respiratory illness symptoms for one child and vaccination cards for 4859 children age 9-59 months.

Recently relocated families had smaller households, less education, less wealth and less knowledge of the local hospital compared with residentially stable families (table 1). Recently relocated families were poorer: 24% earned less than 5000 Bangladeshi taka (US $73) per month compared with 18% of residentially stable families. For the wealth index analysis, the first principal component accounted for 51% of overall variance,

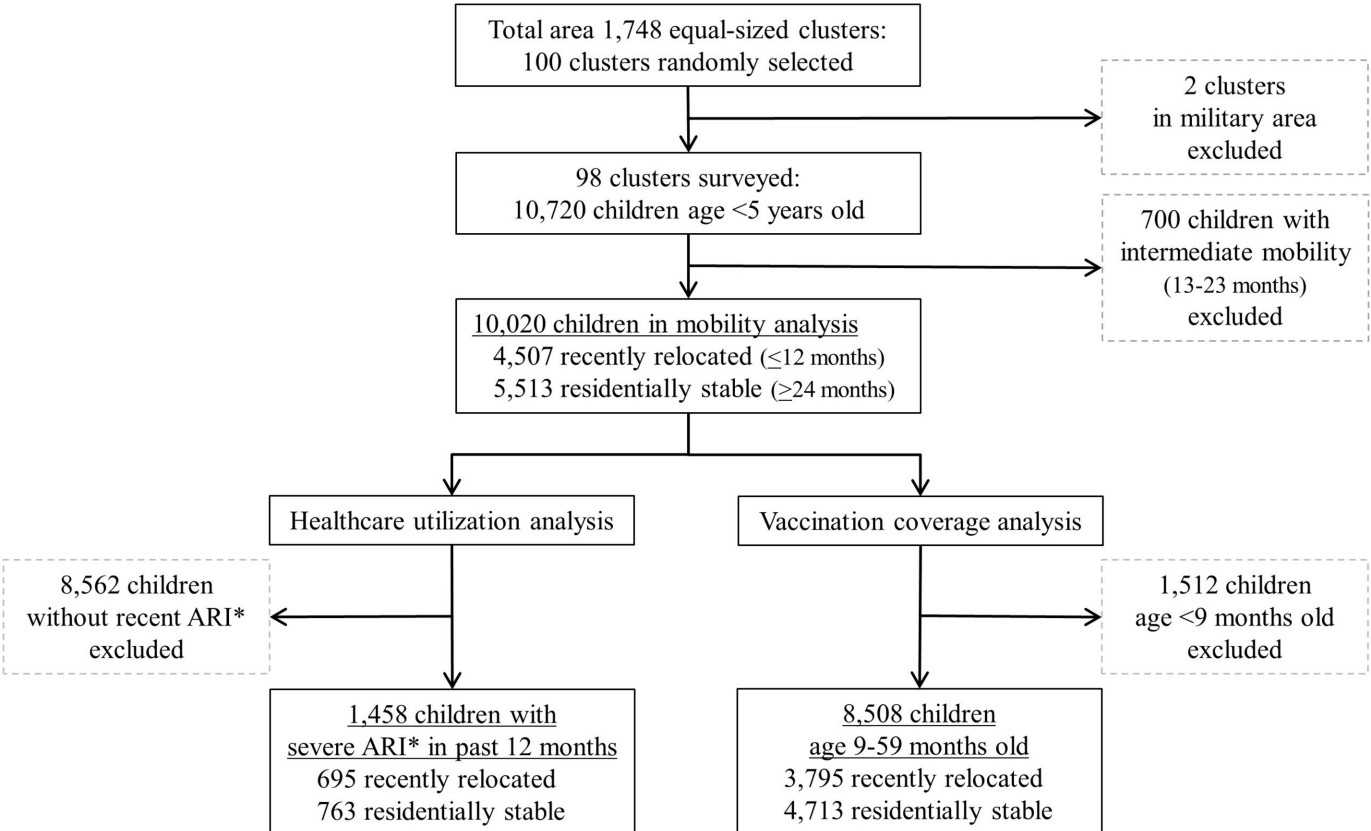

**Figure 1** Children sampled by mobility status for healthcare utilisation and vaccination coverage analysis. *ARI, Acute Respiratory Illness.

with largest contributions from roof and floor materials, sanitation and mobile phone ownership (online supplementary table 1). Among recently relocated families, 48% were in the two lowest wealth quintiles compared with 37% of residentially stable families. Fewer recently relocated caregivers had knowledge of the local hospital, 76%, compared with residentially stable caregivers, 85% ($\chi^2$=142.3, p<0.001). Similar rates of illness in the 12 months prior to survey were reported by all households: 14% to 15% of children with symptoms of severe ARI and 3% to 4% with symptoms of meningitis/encephalitis.

Full vaccination coverage measured by vaccination card plus maternal recall was 83% among all children age 9–59 months (table 2). Full vaccination was 80% among recently relocated children and 85% among residentially stable children (univariate PR 0.94, 95% CI 0.91 to 0.97, p<0.001). Vaccination was lower in households with more children and younger children. Socioeconomic factors, especially mother's education, had the strongest association with vaccination. In multivariable analyses, recently relocated children were 3% less likely than residentially stable children to be fully vaccinated even after adjusting for demographic and socioeconomic factors (multivariable PR 0.97, 95% CI 0.95 to 0.99, p=0.016).

Vaccination was also analysed using only vaccination cards (table 3). At the time of survey, only 43% of all children had vaccination cards available. Fewer recently relocated children had vaccination cards, 36%, compared

with 48% of residentially stable children ($\chi^2$=126.9, p<0.001). Younger children were more likely to have cards than older children: 62% of children 9–23 months old had vaccination cards as compared with 38% of children 24–59 months old ($\chi^2$=505.7, p<0.001). Full vaccination per vaccination card was 83% among recently relocated children and 86% among residentially stable children (univariate PR 0.97, 95% CI 0.93 to 1.00, p=0.083). The 9-59 month age range for vaccination analysis allowed inclusion of a larger sample size of children vulnerable to vaccine-preventable disease. In contrast, routine vaccination schedules focus on children <2 years old and full vaccination coverage per Expanded Programme on Immunisation in Bangladesh and many other countries is defined in children up to 23 months old.[18 33] Narrowing the age range of vaccination analysis to 9–23 months showed similar results although with smaller sample sizes limiting statistical power to detect mobility-vaccination associations (online supplementary table 2 and 3). In addition, using a 10 month age minimum to account for potential delay in measles vaccination recommended at 9 months old showed the same results as a 9 month age cut-off (data not shown). Checking for influential observations with Cook's distances identified no outliers in vaccination analyses (online supplementary figure 1).[34]

Among all children under 5 years old with severe ARI in the past year, 75% visited a qualified medical provider

**Table 1** Mobility status of study households with children age 0-59 months and association with demographic, socioeconomic and health characteristics using t- and $\chi^2$-tests

| | Residentially stable, ≥24 months n=5513 | | Recently relocated, ≤12 months n=4507 | | P value |
|---|---|---|---|---|---|
| **Demographics** | mean | SE | mean | SE | |
| Number of household members | 5.40 | 0.09 | 4.60 | 0.10 | **<0.001** |
| Number of children <5 years in household | 1.28 | 0.01 | 1.25 | 0.01 | 0.194 |
| Age of index child in months | 30.0 | 0.23 | 28.7 | 0.25 | **<0.001** |
| | n | % | n | % | |
| Sex of index child: Male | 2478 | 45 | 2048 | 45 | 0.622 |
| **Socioeconomics** | | | | | |
| Mother's education | | | | | **<0.001** |
| No education | 1133 | 21 | 1162 | 26 | |
| Some schooling | 1142 | 21 | 1176 | 26 | |
| Finished secondary | 1849 | 34 | 1483 | 33 | |
| >Secondary | 1389 | 25 | 686 | 15 | |
| Father's education | | | | | **<0.001** |
| No education | 1261 | 23 | 1139 | 25 | |
| Some schooling | 913 | 17 | 911 | 20 | |
| Finished secondary | 1525 | 28 | 1372 | 30 | |
| >Secondary | 1814 | 33 | 1085 | 24 | |
| Occupation of household head | | | | | **<0.001** |
| Unemployed or other | 482 | 9 | 232 | 5 | |
| Daily labour | 916 | 17 | 1218 | 27 | |
| Shopkeeper or merchant | 1787 | 32 | 1058 | 23 | |
| Salaried service | 2328 | 42 | 1999 | 44 | |
| Monthly household income* | | | | | **<0.001** |
| ≤5000 taka (US $73) | 971 | 18 | 1080 | 24 | |
| 5001–10 000 taka | 1634 | 30 | 1925 | 43 | |
| >10 000 taka (US $145) | 2900 | 53 | 1498 | 33 | |
| Household wealth index† | | | | | **<0.001** |
| Lowest | 1092 | 20 | 1110 | 25 | |
| Second | 922 | 17 | 1021 | 23 | |
| Third | 966 | 18 | 911 | 20 | |
| Fourth | 1455 | 26 | 1155 | 26 | |
| Highest | 1078 | 20 | 310 | 7 | |
| **Health services knowledge** | | | | | |
| Knowledge of local hospital | 4709 | 85 | 3428 | 76 | **<0.001** |
| **Health outcomes** | | | | | |
| Severe acute respiratory illness suffered by index child within 12 months‡ | 763 | 14 | 695 | 15 | **0.026** |
| Meningitis/encephalitis suffered by index child within 12 months§ | 185 | 3 | 164 | 4 | 0.443 |

*12 respondents (eight residentially stable and four recently relocated) did not know or did not disclose household income.
†Polychoric Principal Components Analysis was used to create a household wealth index including structural housing characteristics, cooking fuel, drinking water and sanitation.
‡One residentially stable respondent did not know if child recently had a severe acute respiratory illness.
§Three respondents did not know if child recently had a serious illness with mental status changes (two residentially stable and one recently relocated).

**Table 2** *Using vaccination card plus maternal recall*, vaccination coverage among children age 9–59 months and association with mobility status using univariate and multivariable models with modified Poisson regression

| | Partial vaccination n=1465 (17%) | | Full vaccination* n=7043 (83%) | | PR† | 95% CI | P value |
|---|---|---|---|---|---|---|---|
| **Univariate analyses** | | | | | | | |
| Mobility status | n | % | n | % | | | |
| Residentially stable >24 months | 706 | 15 | 4007 | 85 | Reference | | |
| Recently relocated <12 months | 759 | 20 | 3036 | 80 | **0.94** | **0.91 to 0.97** | **<0.001** |
| Demographics | mean | SE | mean | SE | | | |
| Number of children <5 years in household | 1.31 | 0.02 | 1.25 | 0.01 | **0.96** | **0.93 to 0.98** | **0.002** |
| Age of index child in months | 32.0 | 0.44 | 34.2 | 0.16 | **1.002** | **1.001 to 1.003** | **<0.001** |
| Socioeconomics | | | | | | | |
| Mother's education | n | % | n | % | | | |
| No education | 629 | 31 | 1377 | 69 | Reference | | |
| Some schooling | 401 | 21 | 1540 | 79 | **1.16** | **1.10 to 1.21** | **<0.001** |
| Finished secondary | 323 | 11 | 2513 | 89 | **1.29** | **1.23 to 1.36** | **<0.001** |
| >Higher secondary | 112 | 6 | 1613 | 94 | **1.36** | **1.30 to 1.43** | **<0.001** |
| Occupation of household head | | | | | | | |
| Unemployed or other | 106 | 17 | 505 | 83 | Reference | | |
| Daily labour | 524 | 29 | 1283 | 71 | **0.86** | **0.82 to 0.90** | **<0.001** |
| Shopkeeper or merchant | 348 | 14 | 2084 | 86 | 1.04 | 0.99 to 1.08 | 0.112 |
| Salaried service | 487 | 13 | 3171 | 87 | **1.05** | **1.01 to 1.09** | **0.029** |
| Household wealth status (PCA‡) | | | | | | | |
| Lowest | 596 | 32 | 1290 | 68 | Reference | | |
| Second | 334 | 20 | 1298 | 80 | **1.16** | **1.09 to 1.24** | **<0.001** |
| Third | 242 | 15 | 1348 | 85 | **1.24** | **1.16 to 1.32** | **<0.001** |
| Fourth | 193 | 9 | 2043 | 91 | **1.34** | **1.25 to 1.42** | **<0.001** |
| Highest | 100 | 9 | 1064 | 91 | **1.34** | **1.25 to 1.43** | **<0.001** |
| Health services knowledge | | | | | | | |
| Does *not* have knowledge of local hospital | 354 | 23 | 1218 | 77 | Reference | | |
| Has knowledge of local hospital | 1111 | 16 | 5825 | 84 | **1.08** | **1.05 to 1.12** | **<0.001** |
| **Multivariable analyses with different models** | | | | | | | |
| Mobility, adjusting for demographics (# of children and age of index child) | | | | | **0.94** | **0.92 to 0.97** | **<0.001** |
| Mobility, adjusting for socioeconomics (education, occupation and wealth) | | | | | **0.97** | **0.95 to 0.99** | **0.009** |
| Mobility, adjusting for demographics and socioeconomics | | | | | **0.97** | **0.95 to 0.99** | **0.016** |

*Full vaccination coverage per Expanded Programme on Immunisation includes one dose of BCG, three doses of polio, three doses of diphtheria, pertussis  and tetanus and one dose of measles vaccines.
†PR, Prevalence Ratio.
‡Polychoric Principal Components Analysis was used to create a household wealth index including structural housing characteristics, cooking fuel, drinking water and sanitation.

(table 4). Fewer recently relocated children with severe ARI saw a qualified medical provider, 69%, as compared with 82% of residentially stable children (univariate PR 0.84, 95% CI 0.79 to 0.90, p<0.001). Socioeconomic factors, especially household wealth, were strongly associated with qualified medical provider visits.

Health services knowledge was also strongly associated with acute healthcare visits: 80% of parents who knew about the local hospital sought ARI treatment from a qualified medical provider as compared with 51% of parents who did *not* have knowledge of the local hospital (univariate PR 1.57, 95% CI 1.34 to 1.85, p<0.001). After

**Table 3** *Using vaccination card only*, vaccination coverage among children age 9–59 months who have vaccination cards and association with mobility status using univariate and multivariable models with modified Poisson regression

| | Partial vaccination n=564 (15%) | | Full vaccination* n=3085 (85%) | | PR† | 95% CI | P value |
|---|---|---|---|---|---|---|---|
| **Univariate analyses** | | | | | | | |
| Mobility status | n | % | n | % | | | |
| Residentially stable >24 months | 329 | 14 | 1948 | 86 | Reference | | |
| Recently relocated <12 months | 235 | 17 | 1137 | 83 | 0.97 | 0.93 to 1.00 | 0.083 |
| Demographics | mean | SE | mean | SE | | | |
| Number of children <5 years in household | 1.26 | 0.02 | 1.24 | 0.01 | 0.99 | 0.96 to 1.02 | 0.486 |
| Age of index child in months | 27.1 | 0.64 | 29.6 | 0.25 | **1.002** | **1.001 to 1.003** | **0.001** |
| Socioeconomics | | | | | | | |
| Mother's education | n | % | n | % | | | |
| No education | 165 | 28 | 430 | 72 | Reference | | |
| Some schooling | 123 | 16 | 654 | 84 | **1.16** | **1.09 to 1.24** | **<0.001** |
| Finished secondary | 186 | 14 | 1158 | 86 | **1.19** | **1.11 to 1.28** | **<0.001** |
| >Higher secondary | 90 | 10 | 843 | 90 | **1.25** | **1.17 to 1.33** | **<0.001** |
| Occupation of household head | | | | | | | |
| Unemployed or other | 35 | 13 | 238 | 87 | Reference | | |
| Daily labour | 148 | 26 | 413 | 74 | **0.84** | **0.78 to 0.91** | **<0.001** |
| Shopkeeper or merchant | 143 | 13 | 921 | 87 | 0.99 | 0.94 to 1.04 | 0.778 |
| Salaried service | 238 | 14 | 1513 | 86 | 0.99 | 0.95 to 1.04 | 0.704 |
| Household wealth status (PCA‡) | | | | | | | |
| Lowest | 148 | 27 | 401 | 73 | Reference | | |
| Second | 131 | 21 | 503 | 79 | **1.09** | **1.02 to 1.16** | **0.009** |
| Third | 97 | 15 | 572 | 86 | **1.17** | **1.09 to 1.26** | **<0.001** |
| Fourth | 110 | 10 | 1005 | 90 | **1.23** | **1.15 to 1.32** | **<0.001** |
| Highest | 78 | 11 | 604 | 89 | **1.21** | **1.13 to 1.30** | **<0.001** |
| Health services knowledge | | | | | | | |
| Does *not* have knowledge of local hospital | 101 | 19 | 434 | 81 | Reference | | |
| Has knowledge of local hospital | 463 | 15 | 2651 | 85 | **1.05** | **1.01 to 1.09** | **0.025** |
| **Multivariable analyses with different models** | | | | | | | |
| Mobility, adjusting for demographics (# of children and age of index child) | | | | | 0.97 | 0.94 to 1.01 | 0.126 |
| Mobility, adjusting for socioeconomics (education, occupation and wealth) | | | | | 0.98 | 0.95 to 1.02 | 0.308 |
| Mobility, adjusting for demographics and socioeconomics | | | | | 0.98 | 0.95 to 1.02 | 0.396 |

*Full vaccination coverage per Expanded Programme on Immunisation includes one dose of BCG, three doses of polio, three doses of diphtheria, pertussis and tetanus and one dose of measles vaccines.
†PR, Prevalence Ratio.
‡Polychoric Principal Components Analysis was used to create a household wealth index including structural housing characteristics, cooking fuel, drinking water and sanitation.

adjusting for demographic and socioeconomic factors, recently relocated households were 11% less likely than residentially stable households to visit a qualified medical provider for children with severe ARI (multivariable PR 0.88, 95% CI 0.84 to 0.93, p<0.001). One outlier was identified by Cook's distances in healthcare utilisation analyses (online supplementary figure 1). Excluding this outlier and using robust error variance resulted in similar results to those presented (data not shown).

## DISCUSSION

Recently relocated households were less likely to use both acute and preventive child healthcare services in our study in Dhaka, Bangladesh, and these findings support prior literature exploring the effects of mobility on healthcare utilisation.[6 10–13 16] Household relocation had a strong association with decreased use of qualified medical services for severe ARI. Similarly, household relocation was associated with decreased vaccination rates although this relationship was

**Table 4** Qualified medical provider visits for severe acute respiratory illness within prior year among children <5 years old and association with mobility using univariate and multivariable models with modified Poisson regression

| | No qualified provider for severe ARI* | | Qualified provider for severe ARI* | | PR† | 95% CI | P value |
|---|---|---|---|---|---|---|---|
| | n=358 (25%) | | n=1100 (75%) | | | | |
| **Univariate analyses** | | | | | | | |
| Mobility status | n | % | n | % | | | |
| Residentially stable >24 months | 141 | 18 | 622 | 82 | Reference | | |
| Recently relocated <12 months | 217 | 31 | 478 | 69 | **0.84** | **0.79 to 0.90** | **<0.001** |
| Demographics | mean | SE | mean | SE | | | |
| Number of children <5 years in household | 1.23 | 0.03 | 1.25 | 0.02 | 1.02 | 0.97 to 1.08 | 0.437 |
| Age of index child in months | 29.1 | 0.83 | 23.7 | 0.44 | **0.994** | **0.992 to 0.996** | **<0.001** |
| Socioeconomics | | | | | | | |
| Mother's education | n | % | n | % | | | |
| No education | 134 | 38 | 220 | 62 | Reference | | |
| Some schooling | 127 | 32 | 267 | 68 | 1.09 | 0.96 to 1.23 | 0.170 |
| Finished secondary | 79 | 16 | 425 | 84 | **1.36** | **1.20 to 1.54** | **<0.001** |
| > Higher secondary | 18 | 9 | 188 | 91 | **1.47** | **1.30 to 1.66** | **<0.001** |
| Occupation of household head | | | | | | | |
| Unemployed or other | 24 | 23 | 82 | 77 | Reference | | |
| Daily labour | 139 | 36 | 244 | 64 | **0.82** | **0.72 to 0.94** | **0.005** |
| Shopkeeper or merchant | 89 | 22 | 320 | 78 | 1.01 | 0.90 to 1.14 | 0.848 |
| Salaried service | 106 | 19 | 454 | 81 | 1.05 | 0.93 to 1.18 | 0.435 |
| Household wealth status (PCA‡) | | | | | | | |
| Lowest | 154 | 40 | 228 | 60 | Reference | | |
| Second | 93 | 30 | 220 | 70 | **1.18** | **1.02 to 1.37** | **0.031** |
| Third | 59 | 21 | 220 | 79 | **1.32** | **1.16 to 1.51** | **<0.001** |
| Fourth | 37 | 12 | 277 | 88 | **1.48** | **1.31 to 1.67** | **<0.001** |
| Highest | 15 | 9 | 155 | 91 | **1.53** | **1.35 to 1.73** | **<0.001** |
| Health services knowledge | | | | | | | |
| Does *not* have knowledge of local hospital | 114 | 49 | 118 | 51 | Reference | | |
| Has knowledge of local hospital | 244 | 20 | 982 | 80 | **1.57** | **1.34 to 1.85** | **<0.001** |
| **Multivariable analyses with different models** | | | | | | | |
| Mobility, adjusting for demographics (# of children and age of index child) | | | | | 0.84 | 0.79 to 0.89 | <0.001 |
| Mobility, adjusting for socioeconomics (education, occupation and wealth) | | | | | 0.89 | 0.84 to 0.94 | <0.001 |
| Mobility, adjusting for demographics and socioeconomics | | | | | 0.88 | 0.84 to 0.93 | <0.001 |

*ARI, Acute Respiratory Illness.
†PR, Prevalence Ratio.
‡Polychoric Principal Components Analysis was used to create a household wealth index including structural housing characteristics, cooking fuel, drinking water and sanitation.

less robust. Another key finding was that recently relocated parents were less knowledgeable about the local hospital compared with residentially stable parents and knowledge of the local hospital had as strong an association with acute healthcare visits as some economic factors. Overall, recently relocated children in our study had slightly lower vaccination rates and markedly lower use of acute healthcare services for ARI than residentially stable children.

Study strengths include data focused on urban Bangladesh and exploring vaccination status using different measurements as well as adjusting for socioeconomic factors when examining mobility and health service utilisation. Our study used the Hib impact study's rigorous community surveillance data of households living close to tertiary care paediatric hospitals in Dhaka. Dhaka residents have high mobility and many options for healthcare. Unlike in rural areas, physical access to health services is usually *not* a barrier to healthcare use in urban areas. One study found almost all residents in Dhaka lived within 1 kilometre of primary health services.[35] Routine immunisations are provided free by the government of Bangladesh, but acute care services require out of pocket expenditures, which can be a barrier to access. Our findings on mobility and child health services use in Dhaka could inform health services work in other urban low- and middle-income country contexts.[3 5]

We analysed vaccination using vaccination card data augmented with maternal recall, vaccination card data only and several different age ranges. Accurately measuring vaccinations in children is a known difficulty in public health programmes and research studies, especially in low- and middle-income countries where vaccination cards are frequently *not* available. Some studies have found poor agreement between parental recall, vaccination cards and even official health records.[36–38] By contrast, other studies have found good correlation between maternal report and vaccination cards, and maternal recall is routinely used in Demographic and Health Surveys and Multiple Indicator Cluster Surveys.[39 40] Maternal recall can overestimate or underestimate vaccination history based on education, social desirability bias and vaccine-specific knowledge. Vaccination card retention itself can be affected by parental education, household wealth, age of child and even household relocation. Our data showed more missing cards among recently relocated households and in older children, thus those groups are subject to more maternal recall bias. Using only vaccination cards or narrower age ranges in our vaccination analyses resulted in smaller sample sizes, which limited statistical power, but all analyses showed similar effect estimates of increased mobility associated with decreased vaccination. Moreover, the association between increased household relocation and decreased health services use was still seen even after adjusting for socioeconomic factors known to impact healthcare use.

Study limitations include lack of data on mobility patterns and health services costs. Information on households' prior residences, distances moved or frequency of moving was not available in our dataset. Households moving from rural Bangladesh to urban Dhaka, moving long distances or relocating frequently probably have less knowledge and therefore use of locally available health services.[12] Several studies show that recent rural-to-urban migration is associated with lower vaccination coverage in children.[6 10 16] Lack of data on mobility patterns in our study precludes evaluation of how magnitude of relocation

affected healthcare use. Our findings likely underestimate the negative association between mobility and healthcare utilisation for households with large migration such as rural-to-urban migrants as compared with households with intra-Dhaka relocation where one would expect minimal change in health-seeking behaviour. We were also unable to examine household relocation timing in relation to healthcare use. Recently relocated households were not asked if healthcare visits occurred before or after moving. Healthcare visits before moving would not be relevant to how mobility affects use of health services after moving. Our findings may underestimate the negative association between mobility and health-seeking behaviour because our data included healthcare visits before moving, which were unrelated to knowledge of the new geographic area. Timing of healthcare visits in relation to acquiring knowledge of hospital services was also not available, thus there may be reverse causality of recently relocated households gaining knowledge of local providers after seeking care. Ultimately, our results show a modest overall association between mobility and healthcare use, which could be elucidated by asking about migration patterns including timing of use of health services.

Our study results may not be as generalisable to populations in urban areas without tertiary care hospitals. Our sampling scheme focused on community catchment areas surrounding tertiary care paediatric hospitals. Advantages of this study design were that it allowed examination of healthcare utilisation for severe disease since advanced services were available within a small physical distance. In addition, it was a low-cost way to examine population-level mobility instead of more resource-intensive active surveillance of migrant populations. However, use of health services generally increases with geographic proximity, and studies show this relationship is influenced by many factors including income and slum versus non-slum locations.[35 41] Recently relocated populations may be even more influenced by proximity than residentially stable populations because of fewer socioeconomic resources and lack of knowledge of health services. This would bias our results towards higher rates of health-seeking behaviour among recently relocated households. Recently relocated households in areas without tertiary care services may use health services less because of transport costs and lack of knowledge of health facilities physically distant. In addition, our study included participants who relocated into the study area, but not people who left the study area. In-migrants and out-migrants may be different in their healthcare utilisation patterns, which could also affect generalisability .

Our dataset did not contain cost of services, which is a well-known barrier to healthcare use.[20 21 31] Although vaccinations are provided for free, some non-governmental and private organisations charge fees for patient registration. Even small fees could have lowered vaccination rates. Cost of services, willingness to pay and underlying finances are strongly linked, thus adjusting for socioeconomic factors of parental education and wealth

in our models should have incorporated some cost effects on healthcare use. However, costs could affect recently relocated households disproportionately more than residentially stable households of the same socioeconomic status. One could hypothesise that immediately after relocating, families would first spend money on household goods before preventive medicine fees. Without cost data, we can still conclude from our analysis that increased mobility is associated with decreased healthcare use, but we have limited understanding of mechanisms through which mobility affects healthcare use.

Barriers and delays to using appropriate healthcare services increase mortality.[31 42] One study in India of 290 children hospitalised for pneumonia in a tertiary care centre found that delayed hospital referral, defined as three or more days between symptom onset and hospitalisation, was associated with increased mortality (OR 52.1, 95% CI 6.7 to 402.4, p<0.001) after adjusting for age, residence in slum and illness severity.[42] In this study, incomplete immunisation was also associated with increased mortality (OR 12.3, 95% CI 2.2 to 69.9, p=0.005).[42] Reasons for delayed care-seeking can include access and cost. While cost does influence healthcare use, parents of sick children usually do seek some treatment. In the 2014 Bangladesh Demographic and Health Survey, 52% of urban parents with children with ARI symptoms in the 2 weeks prior to survey sought treatment from a health facility, 23% went to pharmacies and 12% went to traditional practitioners.[43] Only 12% of parents with sick children sought no healthcare treatment at all.[43] While cost does not seem a large barrier to seeking any treatment at all, cost likely influences choice of health provider.

Household relocation disrupts prior relationships with healthcare providers and results in lack of familiarity with local services. Studies show that continuity of care is associated with increased vaccination, fewer emergency department visits and decreased hospitalisation among children.[12 44–46] People usually move to new areas because of pre-existing social connections through family, friends or work.[7 47] These social contacts can act as pathways of important local knowledge, including health services, but recently relocated households have fewer social contacts and access fewer information sources. Other studies have also found that parental attitudes and knowledge are critical factors contributing to use of health services.[6 48–51] One literature review found that practical knowledge about vaccination schedule, timing and logistics had a stronger association with vaccination uptake than scientific knowledge of vaccine names or biologic actions.[50] One study in India of 210 residents in slums and 100 migrant families of construction workers found 28% of slum residents and 64% of migrants identified lack of knowledge of place or time of services as a reason for decreased immunisation.[51] Our study did not ask specifically about knowledge of vaccination services, but future research on knowledge of services could help elucidate how to connect new migrants to care.

Our finding that recently relocated children in Dhaka use fewer qualified health services compared with residentially stable children sheds light on health barriers faced by a growing population of children living in urban centres of low- and middle-income countries. Rigorous community surveillance in hospital catchment areas allows for increased understanding of factors affecting access to and use of healthcare services. Policymakers working to improve urban child health could invest in accurate counting of children living in communities with high household turnover in order to connect recently relocated households to already existing local health services. Further studies by researchers on patterns and mechanisms through which mobility affects healthcare use could inform critical intervention points. Ultimately, cost-effective and targeted interventions to increase appropriate healthcare use among recently relocated children could improve health of future urban populations.

**Acknowledgements** The authors thank study participants, field researchers and collaborative partners of the Hib impact study who included Dhaka Shishu microbiology laboratories and the Johns Hopkins Bloomberg School of Public Health. icddr,b is grateful to the Bangladesh, Canada, Sweden and the United Kingdom governments for providing core/unrestricted support.

**Contributors** SPL was the Principal Investigator and involved in every aspect of the study from conceptualisation to data analysis to manuscript writing and submission. NSK, JA, and LH were involved in data analysis and LH drafted the written work. All authors collaborated on and approved the final manuscript.

**Funding** The original Hib impact study data collection was funded by the Bill and Melinda Gates Foundation and the GAVI Hib Initiative (GR-00580). No additional funding was provided for this study.

**Competing interests** None declared.

**Patient consent for publication** Not required.

**Ethics approval** The Ethical Review Committee of the International Centre for Diarrhoeal Disease Research, Bangladesh (icddr,b) reviewed and approved the study protocol.

**Provenance and peer review** Not commissioned; externally peer reviewed.

**Data sharing statement** Data are available by emailing the corresponding author LH at lhorng@stanford.edu.

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
