## [Reviewer comments · BMJ Open]

ARTICLE DETAILS

TITLE (PROVISIONAL)	Effect of household relocation on child vaccination and health service utilization in Dhaka, Bangladesh: a cross-sectional community survey
AUTHORS	Horng, Lily; Kakoly, Nadira Sultana; Abedin, Jaynal; Luby, Stephen

VERSION 1 - REVIEW

REVIEWER	Dr. Yadlapalli S. Kusuma, Additional Professor All India Institute of Medical Sciences, New Delhi, India
REVIEW RETURNED	06-Sep-2018

GENERAL COMMENTS	This is an important work in the background of ever-increasing migration, particularly of the poor to the cities in search of livelihood. The present manuscript is well prepared and presented well. I would suggest the authors present a conceptual framework highlighting the vulnerability of the low SES, migrant/mobile populations and how this would impact the access to healthcare services. Though the authors could capture the significant differences between the two groups based on their length of stay in Dhaka, they acknowledged that the differences are not robust. They may further discuss the role of rigorous community surveillance of households living close to tertiary care paediatric hospitals on the results of the present study. The authors may also choose to include the details of the vaccination status of the children and treatment-seeking behaviour by households relocation status in the figure.
--

REVIEWER	Tim Crocker-Buque London School of Hygiene and Tropical Medicine
REVIEW RETURNED	09-Sep-2018

GENERAL COMMENTS	Thanks for the opportunity to review this paper, which I read with interest. This is an important topic and this interesting and well-conducted study is worthy of swift publication.
---

	I have several relatively minor comments that could be integrated to improve the manuscript: Page 4, L13: please clarify this sentence 'rapid urbanization is leading to dramatic population growth'. Although the effect is variable in different context, the larger proportion of increase in urban populations is from local births, with a smaller contribution from rural-urban migration, so, if anything, population growth is leading to an increasing urban population. Also, it's much more the migration effect fuelling the increasing slum populations that is resulting in an increasing urban population in slum areas, rather than urbanization fuelling slum growth. Page 4, L19-29: Some of the statistics cited here are relatively out of date, especially given the massive urbanization and slum formation that has taken place in the last 10-15 years. While I am conscious that I am recommending my own paper for inclusion here, it seems extremely relevant, and so I would recommend consideration of the results from Crocker-Buque et al., 2017 Immunization, urbanization and slums - a systematic review of factors and interventions. BMC Public Health. Jun 8;17(1):556. doi: 10.1186/s12889-017-4473-7. It includes a review of cross-sectional studies comparing coverage in urban slum and non-slum areas in a much wider number of contexts than Niger (2006) and Nairobi (2008-2012), including in locations where there is less disparity. Page 6, line 31: is fever + altered mental status a standard definition of a meningitis-type illness? As to my view many children have altered mental status with a significant fever of almost any cause and this is likely to over estimate the number of meningitis cases. Please clarify. Page 6, line 42: this is a useful definition of residential mobility, and is similar to those used in several papers cited in the systematic review described above which also look at coverage in migrants resident for <12 months, which would support the conclusions in the discussion, and would be worthy of consideration for addition. Page 7, lines 5-22: it would be useful to clarify here for the non-vaccination specialist that the EPI schedule would expect children to have had all of these vaccinations before 9 months, and that Hib is contained within the pentavalent vaccine given at 10 and 14 weeks. Page 10: was any knowledge of vaccination providers included as part of the survey? Or just hospital providers? As this would add interesting additional information (did parents know where to be vaccinated and could not attend for other reasons, or did they not know where to go?) Page 11: one of the largest confounding factors here is that highlighted on lines 50-54 here. There are data available from several studies that show significantly lower coverage in children whose parents have migrated from rural to urban areas, and this should be considered here specifically.
--	--

REVIEWER	Adrian Barnett Queensland University of Technology Brisbane
REVIEW RETURNED	12-Dec-2018

GENERAL COMMENTS

This is a challenging area to do a survey in a hard to reach population. The research questions were clear with meaningful endpoints. There was some good discussion of the study's limitations.

I am worried about sampling scheme as it was geared towards those seeking medical attention, as the 100 children who sought help determined the areas that were sampled. So one of the primary outcomes has partly determined the sampling scheme. This is potentially problematic and the authors need to explain why this sampling scheme was used and how it might influence the results both in terms of bias and generalisability.

The design means that only those who move to an area are sampled, not those that move away. Of course those that move to an area have then moved to somewhere, so it may just balance out. However, people moving away may be somewhat different to those moving in. Those moving away are a difficult group to reach, and I think it could be handled simply by acknowledging this potential issue.

The amount of missing data is very important for this survey and needs to be reported in detail for all the key variables. Currently it is simply stated that there were a "large number of missing EPI cards" (page 8). How this missing data could introduce bias should also be considered.

What is modified Poisson regression and how does it differ from standard Poisson regression? Why was it used here? A reference would be handy or a mention of what aspects are modified.

Regression was used but there were no diagnostics, such as residual checks or checks for influential observations. Also the multiple variable regression is labelled as 'multivariate' which is incorrect, see Hidalgo B, Goodman M. Multivariate or multivariable regression? American journal of public health. 2013 Jan;103(1):39–40. Available from: <http://dx.doi.org/10.2105/ajph.2012.300897>

Minor comments

- the abstract uses p-values, but it would be better to use confidence intervals here and in the paper
- I would not use the acronym EPI
- Page 8, say why the 700 children were excluded
- Page 9, say why the second age range of 9 to 23 months was used
- page 10, there may be some reverse causality occurring with the awareness and health-seeking association, as those who went to the hospital may have become aware of the hospital during the illness when they were compelled to seek health information
- page 11, the recall and EPI card question should have been examined using agreement statistics not correlations
- page 12, 'our findings likely underestimate the association', say why
- table 2, the odds ratio for age is very small and this is because it's for a one-month increase in age. It may be sensible to scale the estimate by dividing age by 12 months before running the analysis and hence giving the results in years.

VERSION 1 – AUTHOR RESPONSE

Reviewer #1

Reviewer 1: Comment #1

This is an important work in the background of ever-increasing migration, particularly of the poor to the cities in search of livelihood.

The present manuscript is well prepared and presented well.

I would suggest the authors present a conceptual framework highlighting the vulnerability of the low SES, migrant/mobile populations and how this would impact the access to healthcare services.

Reply: Thank you for the support and suggestion of a conceptual framework on how mobility is related to socioeconomic factors and to access to healthcare services. A conceptual framework by the World Health Organization on social determinants of health has been included in the discussion (page 15, line 24) with citation (reference #43).

Reviewer 1: Comment #2

Though the authors could capture the significant differences between the two groups based on their length of stay in Dhaka, they acknowledged that the differences are not robust. They may further discuss the role of rigorous community surveillance of households living close to tertiary care pediatric hospitals on the results of the present study.

Reply: Thank you for this suggestion, and the role of rigorous community surveillance in hospital catchment areas has been discussed in the conclusion section (revised page 16, line 21)

Reviewer 1: Comment #3

The authors may also choose to include the details of the vaccination status of the children and treatment-seeking behaviour by households relocation status in the figure.

Reply: Thank you for this suggestion. We have revised Figure 1 and incorporated these details.

Reviewer #2

Reviewer 2: Comment #1

Thanks for the opportunity to review this paper, which I read with interest. This is an important topic and this interesting and well-conducted study is worthy of swift publication.

I have several relatively minor comments that could be integrated to improve the manuscript:

Page 4, L13: please clarify this sentence 'rapid urbanization is leading to dramatic population growth'. Although the effect is variable in different context, the larger proportion of increase in urban populations is from local births, with a smaller contribution from rural-urban migration, so, if anything, population growth is leading to an increasing urban population. Also, it's much more the migration effect fuelling the increasing slum populations that is resulting in an increasing urban population in slum areas, rather than urbanization fuelling slum growth.

Reply: Thank you for the comment. The clarification (revised page 4, line 6) and references have been updated.

Reviewer 2: Comment #2

Page 4, L19-29: Some of the statistics cited here are relatively out of date, especially given the massive urbanization and slum formation that has taken place in the last 10-15 years. While I am conscious that I am recommending my own paper for inclusion here, it seems extremely relevant, and so I would recommend consideration of the results from Crocker-Buque et al., 2017 Immunization, urbanization and slums - a systematic review of factors and interventions. BMC Public Health. Jun 8;17(1):556. doi: 10.1186/s12889-017-4473-7. It includes a review of cross-sectional studies comparing coverage in urban slum and non-slum areas in a much wider number of contexts than Niger (2006) and Nairobi (2008-2012), including in locations where there is less disparity.

Reply: Thank you for the suggestion and helpful references. Statistics have been updated in the introduction (revised page 4, line 12 and page 5, line 10) and references have been included.

Reviewer 2: Comment #3

Page 6, line 31: is fever + altered mental status a standard definition of a meningitis-type illness? As to my view many children have altered mental status with a significant fever of almost any cause and this is likely to over estimate the number of meningitis cases. Please clarify.

Reply: Thank you for this question. In the original Hib impact study, surveyed households were asked about any serious illness with acute onset of fever with either convulsions or unconsciousness or altered mental status to define suspected meningoenzephalitis. We have clarified this in the methods (revised page 7, line 4).

Reviewer 2: Comment #4

Page 6, line 42: this is a useful definition of residential mobility, and is similar to those used in several papers cited in the systematic review described above which also look at coverage in migrants resident for <12 months, which would support the conclusions in the discussion, and would be worthy of consideration for addition.

Reply: Thank you for this supporting statement, and we have added this to the methods (revised page 7, line 12).

Reviewer 2: Comment #5

Page 7, lines 5-22: it would be useful to clarify here for the non-vaccination specialist that the EPI schedule would expect children to have had all of these vaccinations before 9 months, and that Hib is contained within the pentavalent vaccine given at 10 and 14 weeks.

Reply: Thank you for these clarifications, and we have added them to the methods (revised page 7, line 24).

Reviewer 2: Comment #6

Page 10: was any knowledge of vaccination providers included as part of the survey? Or just hospital providers? As this would add interesting additional information (did parents know where to be vaccinated and could not attend for other reasons, or did they not know where to go?)

Reply: Thank you for this question. The local hospitals providing acute medical services also provide vaccination services. We asked parents about knowledge of local hospitals in general, but did not specifically ask about knowledge of vaccination services. The question raised about knowledge of vaccination providers is an important area of future research and has been added to the discussion (revised page 16, line 16).

Reviewer 2: Comment #7

Page 11: one of the largest confounding factors here is that highlighted on lines 50-54 here. There are data available from several studies that show significantly lower coverage in children whose parents have migrated from rural to urban areas, and this should be considered here specifically.

Reply: Thank you for this comment, and this important topic has been explored more in the discussion (revised page 13, line 19).

Reviewer #3

Reviewer 3: Comment #1

This is a challenging area to do a survey in a hard to reach population. The research questions were clear with meaningful endpoints. There was some good discussion of the study's limitations.

I am worried about sampling scheme as it was geared towards those seeking medical attention, as the 100 children who sought help determined the areas that were sampled. So one of the primary outcomes has partly determined the sampling scheme. This is potentially problematic and the authors need to explain why this sampling scheme was used and how it might influence the results both in terms of bias and generalisability.

Reply: Thank you for this comment on study design. Our study results may not be as generalizable to populations in urban areas without tertiary care hospitals. Our sampling scheme focused on community catchment areas of tertiary-care hospitals. Advantages of this study design were that it allowed examination of healthcare utilization for severe disease since advanced services were available and physical distance should not have been a barrier to care. In addition, it was a low-cost way to examine population-level mobility instead of more resource-intensive active surveillance tracking migrant populations. However, use of health services generally increases with geographic proximity, and studies show this relationship is influenced by many factors including income and slum versus non-slum locations.(references #34 and 41) Recently relocated populations may be even more influenced by proximity than residentially stable populations because of fewer socioeconomic resources and lack of knowledge of health services. This would bias our results towards higher rates of health-seeking behavior among recently relocated households. Recently relocated households in other geographic areas without tertiary care services may use health services less because of the cost of transport and the lack of knowledge of health facilities physically distant. This discussion has been added to study limitations (revised page 14, line 8) with citations (references #34 and 40).

Reviewer 3: Comment #2

The design means that only those who move to an area are sampled, not those that move away. Of course those that move to an area have then moved to somewhere, so it may just balance out. However, people moving away may be somewhat different to those moving in. Those moving away are a difficult group to reach, and I think it could be handled simply by acknowledging this potential issue.

Reply: Thank you for this comment, and we agree that in-migrants may differ from out-migrants in their healthcare utilization patterns, which could also affect generalizability. This discussion point has been added to the study limitations section (revised page 14, line 22).

Reviewer 3: Comment #3

The amount of missing data is very important for this survey and needs to be reported in detail for all the key variables. Currently it is simply stated that there were a “large number of missing EPI cards” (page 8). How this missing data could introduce bias should also be considered.

Reply: Thank you for this comment. Missing data had already been reported in different areas of the results section and in notes under Table 1. We moved the details of missing data to the first paragraph of the results section to make this information more organized and noticeable (revised page 9, line 23). The group of older children had more missing vaccination cards than younger children, as routine vaccination schedules focus on children <2 years old. More recently relocated households had missing cards than residentially stable households. This detail on missing cards is added to the results section (revised page 10, line 24), and bias from missing vaccination cards is reviewed in the discussion section (revised page 12, line 24).

Reviewer 3: Comment #4

What is modified Poisson regression and how does it differ from standard Poisson regression? Why was it used here? A reference would be handy or a mention of what aspects are modified.

Reply: Thank you for these questions. The modified Poisson regression uses a robust error variance or sandwich estimator. In binomial data with common outcomes, standard Poisson regression overestimates variance. Robust Poisson regression corrects the overestimated error. We chose modified Poisson regression to model prevalence ratios for common binary outcomes because logistic regression is more applicable to rare outcomes and because log-binomial regression models may fail to converge.

The following two references and clarification have been added to the methods (revised page 8, line 17; references #27 and 28):

-Zou G. A modified poisson regression approach to prospective studies with binary data. *Am J Epidemiol* 2004;159:702–6.

-Chen W, Shi J, Qian L, et al. Comparison of robustness to outliers between robust poisson models and log-binomial models when estimating relative risks for common binary outcomes: a simulation study. *BMC Med Res Methodol* 2014;14:82. doi:10.1186/1471-2288-14-82

Reviewer 3: Comment #5

Regression was used but there were no diagnostics, such as residual checks or checks for influential observations.

Reply: Thank you for this comment. Regression diagnostics included checks for influential observations with Cook’s distance calculations. Residual plots were difficult to interpret because our model had binary outcomes. Cook’s distances revealed no outliers in the vaccination analyses and only one outlier in healthcare utilization analyses. Excluding this one outlier and using robust error variance resulted in very similar results to those presented in Table 4. This information has been added to the methods (revised page 9, line 1) and results section (revised page 11, lines 10 and 23) as well as an additional Supplemental Figure 1.

Reviewer 3: Comment #6

Also the multiple variable regression is labelled as 'multivariate' which is incorrect, see Hidalgo B, Goodman M. Multivariate or multivariable regression? *American journal of public health*. 2013 Jan;103(1):39–40. Available from: <http://dx.doi.org/10.2105/ajph.2012.300897>

Reply: Thank you for this clarification. The correct term of multivariable regression has been changed throughout the manuscript and this reference has been added.

Reviewer 3: Comments #7-15

Minor comments

- the abstract uses p-values, but it would be better to use confidence intervals here and in the paper
- I would not use the acronym EPI
- Page 8, say why the 700 children were excluded
- Page 9, say why the second age range of 9 to 23 months was used
- page 10, there may be some reverse causality occurring with the awareness and health-seeking association, as those who went to the hospital may have become aware of the hospital during the illness when they were compelled to seek health information
- page 11, the recall and EPI card question should have been examined using agreement statistics not correlations
- page 12, 'our findings likely underestimate the association', say why
- table 2, the odds ratio for age is very small and this is because it's for a one-month increase in age. It may be sensible to scale the estimate by dividing age by 12 months before running the analysis and hence giving the results in years.

Reply:

Thank you for all the detailed review and many thoughtful comments. Our responses are below:

- The corresponding χ^2 -test statistics and 95% CIs have been added to p-values throughout the manuscript.
- EPI acronym has been removed throughout the manuscript.
- The 700 children excluded were those living in their current residence 13-23 months and classified as intermediately mobile. This has been clarified in the results (revised page 9, line 18).
- The 9-23 month age range was used because full vaccination coverage per Expanded Programme on Immunization in Bangladesh and many other countries is evaluated in children up to 23 months old. This has been clarified in the results (revised page 11, line 3)
- Reverse causality: This discussion point has been added to the study limitations section (revised page 14, line 4).
- Maternal recall and EPI/vaccination card question: Accurately measuring vaccinations in children is a known difficulty in public health programs and research studies. Some studies have found poor agreement between parental recall, vaccination cards, and even official health records. By contrast, other studies have found good correlation between maternal report and vaccination cards, although mothers can overestimate or underestimate vaccination history based on education, social desirability bias, and knowledge of vaccines. We chose to augment vaccination card data with maternal recall instead of comparing data sources, an approach used in many surveillance studies of low- and middle-income countries including Demographic and Health Surveys and Multiple Indicator Cluster Surveys. We have expanded our discussion and references on this topic in the discussion section (revised page 12, line 24).
- The phrase 'our findings underestimate' was used twice and both instances have been explained in more detail in the study limitations section (revised page 13, line 21 and line 27).

-Age calculations: Thank you for this suggestion, however, we have kept age in months because this seems a more meaningful increment when discussing vaccinations in young children.

VERSION 2 – REVIEW

REVIEWER	Dr. Y. S. Kusuma All India Institute of Medical Sciences, New Delhi, India
REVIEW RETURNED	28-Jan-2019

GENERAL COMMENTS	Thank you for the opportunity to read this important work. I have minor suggestions. I suggest incorporating the conceptual framework in the Introduction. Regarding Supplemental Table 1, suggest checking the walls classification. Tin walls are mentioned in finished walls category. Pl. refer to https://dhsprogram.com/pubs/pdf/AS61/AS61.pdf . Regarding Household wealth status, the authors may choose to replace 'richest' by 'Upper'.
---

REVIEWER	Tim Crocker-Buque Health Protection Research Unit in Immunisation London School of Hygiene and Tropical Medicine London, UK
REVIEW RETURNED	27-Jan-2019

GENERAL COMMENTS	No further comments. The manuscript is suitable for publication.
--

REVIEWER	Adrian Barnett Queensland University of Technology Australia
REVIEW RETURNED	24-Jan-2019

GENERAL COMMENTS	No further comments, the authors have answered all my previous questions.
---

VERSION 2 – AUTHOR RESPONSE

Reviewer #3

No further comments, the authors have answered all my previous questions.

Reply: Thank you for your previous comments and review.

Reviewer #2

No further comments. The manuscript is suitable for publication.

Reply: Thank you for your previous comments and support.

Reviewer #1

Reviewer 1: Comment #1

Thank you for the opportunity to read this important work. I have minor suggestions.

I suggest incorporating the conceptual framework in the Introduction.

Reply: Thank you for this suggestion. We have incorporated the conceptual framework in the introduction section (page 5, line 14).

Reviewer 1: Comment #2

Regarding Supplemental Table 1, suggest checking the walls classification. Tin walls are mentioned in finished walls category. Pl. refer to <https://dhsprogram.com/pubs/pdf/AS61/AS61.pdf>.

Reply: Thank you for your careful review. Per your suggestion, we re-examined classifications for wall materials from our dataset, Demographic Health Surveys (DHS), as well as UNICEF Multiple Indicator Cluster Surveys (MICS). We found that tin has been classified in different ways across all these surveys. Bangladesh 2007, 2011, and 2014 DHS classify tin walls as “finished”. By contrast, the reference above is for 2016 DHS on malaria in sub-Saharan Africa which classifies walls made of tin/cardboard/paper/bags as “natural”. Bangladesh 2006 MICS classifies tin sheet walls as “rudimentary”, however Bangladesh 2012 MICS lists tin walls separately and in between “rudimentary” and “finished”. We used a community survey dataset from Bangladesh in which tin walls were classified as “finished” which is consistent with Bangladesh DHS surveys.

This variability in housing classification reflects the overall difficulty in accurately estimating wealth using material goods. We ultimately chose to include mother’s education and household head occupation in addition to household wealth as socioeconomic factors in our final analyses.

Reviewer 1: Comment #3

Regarding Household wealth status, the authors may choose to replace 'richest' by 'Upper'.

Reply: Thank you for this comment on language choice. We have chosen to replace wealth quintile labels with the same language used in Bangladesh DHS surveys: “lowest”, “second”, “third”, “fourth”, and “highest”.

We have updated files and reference numbering according to the changes mentioned above. We have

uploaded highlighted and clean copies of the revised manuscript as instructed. Thank you very much for your kind consideration for publication.